# UV LED Curable Perfluoropolyether (PFPE)-Urethane Methacrylate Transparent Coatings for Photonic Applications: Synthesis and Characterization

**DOI:** 10.3390/polym15142983

**Published:** 2023-07-08

**Authors:** Christian Dreyer, Dana Luca Motoc, Mathias Koehler, Leonid Goldenberg

**Affiliations:** 1Department Fiber Composite Material Technologies, Faculty Engineering and Natural Sciences, Technical University of Applied Sciences Wildau, Hochschulring 1, 15745 Wildau, Germany; christian.dreyer@th-wildau.de (C.D.); leonid.goldenberg@th-wildau.de (L.G.); 2Research Division Polymeric Materials and Composites PYCO, Fraunhofer Institute for Applied Polymer Research IAP, Schmiedestr. 5, 15745 Wildau, Germany; mathias.koehler@iap.fraunhofer.de; 3Department of Automotive and Transport Engineering, Faculty of Mechanical Engineering, Transilvania University of Brașov, 29 Eroilor Av., 500036 Brașov, Romania

**Keywords:** perfluoropolyether (PFPE), fluoroacrylates, UV LED curable coating, viscoelastic properties, refractive index, thermo-optic coefficient, optical loss, dielectric properties

## Abstract

The contribution aims to bring forth a novel synthesis route in developing transparent UV LED-curable coatings accounting for various exposure options. A selection of perfluoropolyether (PFPE)-urethane methacrylate and acrylate resins, free-radical photo-initiator Omnirad 2100, and two distinct silane-based crosslinking agents were blended under a weight ratio of 75:20:5 (without crosslinker) and 70:15:5:10, respectively. The coatings were cured under a UV LED 4 × 3 matrix light emitting source, in a chamber under a controlled atmosphere, by means of an in-house developed conveyor belt type platform, at different conveyor belt speeds (5, 50, 150, 250, and 500 mm/s). The morphologies of fabricated coatings were characterized by FTIR revealing high conversion rates (e.g., from 98 to 100%) for increased exposure time as a result of the 5 or 50 mm/s values, on all combinations. Dynamic–mechanical and optical properties of UV LED-cured transparent coatings were also investigated. A negative shift of the glass transition temperature values with a decrease in exposure time, in all combinations, from about 60 °C to 30 °C, along with storage moduli lowering in the glassy plateau further favors higher exposure times for curing. The refractive indices of poly-mers were from 1.38 to 1.40, whereas the thermo-optic coefficients are showing minor changes around the value of 2.55∙10^−4^ K^−1^.

## 1. Introduction

The development of functional and sustainable high-performance coatings based on UV-curable fluorinated resins captured the attention of the scientific community over the past few years, as communication technologies and related devices and/or components are pushing forward increasingly the requirements on their optical properties. Owing to their unique properties ranging from high thermal and chemical resistance [1,2,3], low surface tension [4], low refractive index and transmittance losses [5,6], large thermo-optic coefficients [7], and color stability [8], UV-curable fluorinated resins are the preferred selection in several industrial and commercial applications. In addition to integrated optical devices, cladding for optical fibers, photovoltaics, and microelectromechanical system (MEMS) devices, other applications such as marine antifouling coatings [9,10], automotive clear coat formulations [11,12] and batteries [13,14], coatings of biomedical devices (e.g., artificial hearts, vascular prostheses or catheters) [15], microfluidic devices [16,17], textile industry, food packaging [18], etc., were also suggested.

One of the most widely deployed and researched UV-curable perfluoropolyether (PFPE)-urethane methacrylate resin, referred to as Fluorolink^®^, developed by Solvay SA under its top-tier portfolio of specialty polymers, can be readily blended with conventional acrylate monomers. This finds an application niche in the aforementioned areas. In relation to this, Ursino et al. [19] reported on the use of Fluorolink^®^ MD700 as a surface modifier for polyamide (PA) and polyether sulfone (PES) to deliver hydrophilic/hydrophobic membranes for water treatment showing increased stability and resistance to chemical cleaning agents and salty solutions [19]. Gibbons et al. [20] co-authored an extended database containing experimentally measured refractive indices, optical transmission, and optical absorption spectra of the same oligomer prepared as thin film by variable spectroscopic ellipsometry, as an open access content to be used for various photo-curable synthesis processes [20]. Bongiovanni et al. [21] reported on Fluorolink^®^ D10H [21,22] and Fluorolink^®^ MD700 [23] considering various synthesis routes, a wide range of commercial photo-initiators (e.g., Darocur^®^ and Irgacure^®^ representatives), identical UV exposure conditions. The resulting materials were characterized by mechanical, thermal, or surface properties and are relatively highly referred to in the literature. Thus, in addition to reporting enhanced thermal resistance, reasonable mechanical and chemical resistance, very low refractive index, and surface tension, the authors debated on extending the application range as adhesives. Recently, Goralczyk et al. [24] underlined the versatility of Fluorolink^®^ MD700, both from application and manufacturing technology perspectives, by 3D printing a transparent microfluidic device using stereolithography (SLA) [24]. Their results exhibited enhanced mechanical, thermal, and optical properties owing to the inherent features of PFPE polymer.

Radical type photo-initiators are equally important in carrying out the photopolymerization processes, particularly in industrial environments, since their addition to the chemical solution leads to higher production rates, diminished energy consumption and environmental pollution while their amount together with curing conditions significantly influence obtained material properties [25]. The selection of photo-initiators depends mainly on its compatibility with the resin. Omnirad free radical photo-initiators of type I from IGM Resins^®^, former known as the Irgacure series, are the most common photo-initiators, followed by Darocur^®^ TPO and Lucirin^®^ TPO from BASF [26].

To further underscore, crosslinking strategies and curing process play a crucial role both for coating morphology, crosslinking degree, and material properties. In their state-of-the-art contribution to various curing technologies, Javadi et al. [27] revised, *inter-alia*, the UV curing line equipment, from lamps and reflectors to LED-based light sources, applications related configuration adaptation, environmental conditions, a range of polymers selection and photoinitiated mechanism of their polymerization from chosen references [27]. Kredel et al. [28] reported different crosslinking strategies for fluorine-containing polymer coatings cured by UV mercury lamps [28]. Reported results demonstrated enhanced water and oil repellency along with the durability of developed coatings. In their contribution, Ghazali et al. [29] favored UV LED technology deployment for polymer-based coatings curing. They proved energy efficiency, reduced heat generation, easiness during operation, and environmental safety compared to UV mercury lamps, respectively [29]. Further, they emphasized the limited number of publications on polyurethane coatings using UV LED and demonstrated the enhanced properties of their formulation, from the high degree of C=C conversion (96–98%) resulted in relatively high transparency and development of the green side aspect of photopolymerization. Similar statements can be found in the recent review of Dall Agnol et al. [30] on UV-curable waterborne polyurethane coatings [30]. The authors pointed out the main factors that determine the UV curing behavior of UV-curable coatings, including the wavelength and intensity of radiation, reaction temperature, and reactivity of functional groups. In relation to the above-mentioned, increased radiation exposure time of formulations lead to a decrease in the crosslinking rate of the C=C bond and further degradation of other material properties, including tensile strength, hardness, or solvent resistance.

Owing to their low refractive index values, fluorinated polymers are sought as the best candidates for developing optical structures as antireflecting, protective, or barrier coatings [31]. Furthermore, thermo-optic coefficients (TOC) are of critical importance as small changes may affect device performances [32,33]. Razali et al. reported on thermo-optical coefficients of two different acrylate, cyclomer, and fluorinated polymers, respectively. Their findings revealed a linear dependency of refractive index as a function of temperature, with slopes of −10^−4^, the lowest being found for fluorinated polymers.

The main objective of this study was to provide a new synthesis route for a selection of fluoroacrylate oligomer and monomer for the fabrication of transparent coatings for photonic applications of enhanced performance, with the aid of an in-house developed UV LED computer-assisted curing system. The UV LED technology was formerly used within the research group on a selection of fluorinated resins owing to their application potential for integrated optical devices [34]. In summary, it was concluded that exposure time strongly influences the conversion rate and hence the network density, which is significant in the coating properties. However, other combinations had to be exploited to provide new low refractive index cladding materials for optical glass fibers that can be cured under conditions similar to industrial environments. Application range as optical and photonic devices covers further optical coatings, lens coatings, and light guides since cost-effectiveness and environmental friendliness are considered in the design requirements along with high-performance material properties.

The main contributions underpinning this study are summarized as follows:A novel synthesis route for transparent coatings preparation based on selection (e.g., different crosslinking agents), and curing conditions (e.g., exposure time) with high conversion rates over extended UV LED exposure time range.A selection of fluorinated resin-based transparent coatings deliverables under different curing exposure times.Data processing and analysis on the comprehensive database containing viscoelastic and optical properties (i.e., refractive indices, thermo-optical coefficients, and optical losses).

## 2. Materials and Methods

### 2.1. Materials

A combination of a bifunctional perfluoropolyether (PFPE)-urethane methacrylate oligomer (Fluorolink^®^ MD700) with a high fluorine content (62 wt.%) and a tetra-functional PFPE-urethane acrylate (Fluorolink^®^ AD1700, both Solvay Specialty Polymers) with a ratio of 80:20 wt.%, was stirred at 100 °C for 4 h to act as reference polymer blend. The mixture was left to cool down to room temperature prior addition of supplementary components. The oligomer in the above combination was chosen for its high reactivity and low viscosity (430 cP at 25 °C) as well as water/oil repellence properties whilst the second one was added to the acrylic mixture owing to its effectiveness on fingerprints and stains removal.

The photo-initiator used in all formulations is a phenyl bis(2,4,6-trimethylbenzoyl)-phosphine oxide (Omnirad^®^ 2100, IGM Resins B.V., Waalwijk, The Netherlands) due to its efficiency to initiate radical polymerization of acrylates after UV exposure. This was added to the resin formulation above (i.e., resulting weight ratio 75:20:5), 5 wt.%, stirred at room temperature for 15 min and stored at 5 °C in dark conditions for further usage.

Two crosslinking agents, namely, dimethyl(divinyl)silane (H_2_C=CH)_2_Si(CH3)_2_, 97% purity (DM(DV)S) and trimethoxy(vinyl)silane H_2_C=CHSi(OCH_3_)_3_, 97% purity (TM(V)S) (both Sigma-Aldrich Chemie GmbH, Taufkirchen, Germany) was added separately to the basic formulation above, in a weight ratio of 70:15:5:10. The structures of Fluorolink components were specified using 1H, 13C, and 19F NMR spectroscopy registered using Spinsolve 80 NMR spectrometer (Magritek, Wellington, New Zealand and Aachen, Germany) being in line with other traceable references (see [35,36]): Fluorolink MD700, 1H NMR (CDCl3, 80 MHz) δ: 6.1 and 5.55 (CH2=C of methacrylate group), 5.87 (NH of urethane group); 13C NMR (CDCl_3_, 80 MHz) and Fluorolink AD1700, 1H (CDCl_3_, 80 MHz) δ. Figure 1 depicts the molecular structures of components according to data of manufacturers and data of NMR spectra.

### 2.2. Coating Preparation and UV LED-Assisted Curing Process

The fluorinated acrylate films were prepared on standard BK7 soda-lime glass slides (dimension L 76 × H 26 mm, thickness 1.2–1.5 mm) (from Carl Roth GmbH, Karlsruhe, Germany) by doctor blading technique at room temperature resulting in 300 µm thick films. These were flushed into a mold for 2 min under nitrogen atmosphere and subjected to UV LED curing by passing under a conveyor belt-type platform at different speeds (5, 50, 150, 250, and 500 mm/s). A UV crosslinking reaction was carried out under nitrogen atmosphere to avoid oxygen inhibition, at T = 23 ± 2 °C. The UV LED source, disposed of as a matrix of 4 × 3 LEDs emitting at 390 nm radiation of 705 mW/cm^2^, was kept at 12 mm distance from the coating surface.

Samples for measurement of optical properties, such as thermo-optic coefficients (TOC) and optical loss, were spin-coated using a WS 400-6TFM LITE spin-coater (from Laurell Technologies Co., Lansdale, PA, USA). Film thickness resulted in approx. 20 µm through spinning at 1500 rpm for 10 s. Next, these were subjected to UV-curing for 15 min in a UVH-254 lamp (from Panacol-Elosol GmbH, Steinbach (Taunus), Germany) keeping the same values for radiation and distance from their surface as above mentioned. No additional baking was necessary to complete the crosslinking reaction. Thin films were subjected to experimental measurements promptly after their preparation to avoid post-curing effects.

### 2.3. Curing Behavior and Characterization

The chemical components of the UV LED cured fluorinated acrylate film samples were confirmed by real-time Fourier transform infrared (FT-IR) spectroscopy using a Nicolet 5700 spectrometer (Thermo-Fisher Scientific Inc., Waltham, MA, USA), equipped with a diamond probe, in the 4000–600 cm^−1^ range. The FTIR spectra of samples, both upper and bottom sides, were recorded in the 3000–150 cm^−1^ range, 32 scans, resolution of 4 cm^−1^. The degree of curing was evaluated by monitoring the intensity of the peak at 1640 cm^−1^ assigned to the C=C bond stretch vibrations in the methacrylate group. Omnic™7.2 software was used for spectra evaluation. Degree of conversion (%*Conversion*), demonstrating the curing efficiency, was further evaluated using the following formula:(1)$$\%Conversion=\left(1-\frac{[I{]}_{1640}^{c}}{[I{]}_{1640}^{u}}\right)\cdot 100$$
where $[I{]}_{1640}^{c}$ and $[I{]}_{1640}^{u}$ are the absorbances at 1640 cm^−1^ of fluorinated acrylate coatings before and after the exposure to UV LED emission at different times, as set by the imposed speed of the carriage return holding device. Excerpts of the FT-IR spectra are shown in Figure 1 on all selections, as retrieved at a curing rate of 5 mm/s. Figure 1 demonstrates complete curing of the sample though total thickness as spectra taken from both sides shows small differences.

### 2.4. Viscoelastic Properties by Means of Dynamic Mechanical Analysis (DMA)

Dynamic mechanical measurements were performed using a controlled stress rheometer RSA6 analyzer from Rheometric Scientific (TA Instruments, New Castle, DE, USA) at a frequency of 1 Hz, in tensile mode, from −40 °C to 250 °C, using a 5 K/min heating rate in the dynamic step. The experimental runs comply with the ASTM D5026-15 standard procedure or its ISO 6721-4:2019 equivalent. Sample dimensions were set as 55 × 11.5 × h mm^2^ (length × width × height) and carefully shaped using a sharp cutter. Measurement set-up and data acquisition were handled using a Rheology Advantage Instrument Control AR environment enabling information gathering for the storage (E’) and loss (E”) moduli as well as loss factor (tan δ) as a function of temperature.

### 2.5. Optical Properties Measurements

#### 2.5.1. Refractive Index (*n*)

The refractive index (*n*) of fluorinated acrylate films was determined using a Metricon 2010/M prism coupling system (Metricon Co, Pennington, NJ, USA) in TE mode, using a wavelength of 632.8 nm under a pressure of 22 psi, at 20 °C. The refractive indices, as obtained at room temperature, account for the various degree of curing as the UV LED head moves at different speeds. Free-standing films removed from substrate were used for this measurement. The values of refractive index were obtained from the kick at measurement curve using software in Metricon 2010/M.

#### 2.5.2. Thermo-Optic Coefficients (TOC)

Thermo-optic coefficients (TOC) were determined from the thermal dependence of the refractive index (*n(T)*). Samples were prepared in this case on Si wafer substrate and increasing the temperature gradually, within the range of 30–200 °C, TOC values were obtained as slope of *n(T)* curve.

#### 2.5.3. Optical Loss

Optical loss values of spin-coated UV-cured films were acquired using the prism coupling system Metricon 2010/M as above. The samples were prepared on long Si substrate (at least 6 cm long). First, the films of Fluorolink^®^ MD700 were prepared and cured, then the films of investigated samples were deposited on the top and cured. Optical losses of the planar waveguides were measured by a technique involving measurement of transmitted and scattered light intensity as a function of propagation distance along the waveguide at 633 nm and 1547 nm [37]. The values of optical losses were calculated in Metricon software by exponential fitting of the scattered light intensity vs. distance curve.

## 3. Results and Discussions

### 3.1. Structural Characterization of UV LED Cured Coatings

Variation of exposure times for the developed fluorinated acrylates in the absence/presence of silane-based crosslinking systems lead to different curing degrees, either on glass or free sides, as shown in Figure 2a,b. Thus, at high exposure rates, i.e., low curing times, the double bond of end-capping groups is not entirely available, whilst for small exposure rates (i.e., 5 or 50 mm/s) the curing degree remains almost unchanged [27]. The addition of dimethyl(divinyl)silane into the formulation greatly enhances the conversion rate, i.e., the crosslinking rate of the C=C bond, either favoring or compromising their dynamic–mechanical or optical properties, as shown in the following sections. 

In addition, it should be noted that no sol–gel reaction occurs with the combinations described herein. A combination of silanes with double bonds is used as crosslinking agents, and not as sol–gel precursors. Consequently, crosslinking occurs via free radical double-bound polymerization between the vinyl group of silanes and the acrylate group in the oligomer. Since the silane concentration is relatively low (about 10%) and there is no steam treatment to promote the sol–gel reaction, proven the preparation of formulation in a closed vessel leads to the non-existence of conditions for hydrolysis. Moreover, film preparation and disposal are relatively quick, and UV LED curing takes place in a controlled environment allowing brief exposure to air.

### 3.2. DMA Properties of UV LED-Cured Coatings

The storage modulus (E’) and loss tangent (tan δ) variation with temperature of fluorinated acrylate base film samples at various exposure rates are shown in Figure 3a,b. High curing time or small UV LED exposure rates (i.e., 5 and 50 mm/s) reveal a linear evolution of the E’ in their glassy plateau region (−40 °C—approx. 25 °C) followed by a sharp decrease due to the glass transition of the polymer mixture. As the films are exposed to equally dosed UV LED light (rate values > 150 mm/s), they soften, and their storage modulus tends to decrease. The α-glass transition temperatures (T_g_) range from 27 °C to 60 °C, the highest corresponding to the highest UV LED exposure time.

Figure 4 further shows the influence of the silane-based crosslinking system on the elastic properties of fully cured coatings. In the presence of TM(V)S, the storage modulus of the coating experiences a degradation in the glassy plateau and a negative shift of the T_g_ peak by about 20 °C compared to the reference. On the other hand, accounting for identical curing conditions, in the presence of the DM(DV)S, the storage modulus and the T_g_ peak reveal relatively small to almost negligible variations with respect to the selected reference.

The glass transition temperatures, as identified from the maximum of α transition peaks in tan δ curves, are summarized in Table 1 along with the storage modulus of crosslinked films as obtained at 25 °C. Viscoelastic properties of cured films, in the presence of silane-based crosslinker systems, for higher exposure rate values (>150 mm/s) are not reported as they experience sudden fragmentation under loading conditions. The PFPE-U(MA/A) reference films, as cured at 250 mm/s and 500 mm/s rates were further revealing a decreasing tendency on their glass transition T_g_ values, to 39.5 °C and 30.5 °C, respectively. This shift in glass transition temperatures toward lower values is a direct effect of the degree of polymerization of fluorinated acrylates under different exposure times. Further, this must be viewed in correlation with their degree of conversion, as reported in the above section, as for a nearly completely converted sample, no further polymerization occurs. To the latter, the content of PFPE plays a major role as it has an adverse effect on the curing reaction and contribute significantly to the diminishing of the coating stiffness [4]. All samples exhibited a single *T_g_* peak suggesting a good miscibility between the hard and soft segments.

Table 1 also summarizes the crosslinking densities of UV LED cured fluorinated acrylates derived from the minimum value of storage modulus in the rubbery plateau divided to correspondingly temperature and gas constant [38]. As foreseen, crosslinking densities are highly influenced by the UV exposure rate and composition of silane-based crosslinker deployed.

### 3.3. Optical Properties of UV LED Cured Coatings

#### 3.3.1. Refractive Index (n)

Changes of the refractive indices of tailored fluorinated acrylate coatings as a function of selected exposure speed for the UV LED head used on their curing are exhibited in Figure 5. Thus, the lowest translational rates (e.g., 5 and 50 mm/s), consequently higher time for curing, exhibited refractive index variation in the 1.389–1.402 range. As it can be traced, lower refractive index values correlate with lower crosslinking density values since we have a higher concentration of unreacted oligomers, and apparently oligomers have a lower refractive index compared to crosslinked polymers.

Further insight into the refractive indices of fluorinated acrylates and their dependence on the crosslinking degree under different processing conditions can be traced in the correlation matrices in Appendix A, respectively. As can be seen, there is a strong linear relationship between the degree of cure of fluorinated polymers and their refractive indices, considering both the glass and free side. The reported Pearson correlation coefficients are all positive and show higher values for polymer combinations containing crosslinkers (free side—ca. 0.920 in the presence of DM(DV)S and TM(V)S crosslinkers; glass side—0.945 in the presence of DM(DV)S and 0.962 in the presence of TM(V)S). As for the relationship between the different exposure rates under UV emission and the measured refractive indices for selected combinations, the correlation coefficients have a negative sign, showing a decreasing tendency of the latter as the former increases. 

#### 3.3.2. Thermo-Optic Coefficients (TOC)

Figure 6 reveals the temperature dependence of the refractive index (*n(T)*) for all fluorinated acrylate films, TE mode at 633 nm. The figure also indicates the linear dependence of the refractive index as a function of temperature. Thus, this dependence relates the refractive index at room temperature (*n*_0_) and its value at a higher temperature, in the form:(2)$$n\left(T\right)=n_{0}+\frac{dn}{dT}\left(T-T_{0}\right),$$
where *dn/dT* is the thermo-optic coefficient (TOC) identified as the curve slope, *T*_0_ is the room temperature. Different crosslinker systems produce differences in the TOC values as compared with the reference fluorinated acrylate polymer resin. Thus, dimethyl(divinyl)silane addition brings no changes in the thermo-optic coefficient −2.57∙10^−4^ K^−1^ as compared to the reference, whereas the addition of trimethoxy(vinyl)silane shifted TOC towards −2.5∙10^−4^ K^−1^. Glass transition was not manifested in TOC curves probably due to low value of *T_g_* for these materials and measurement would require of cooling the sample, which is not possible in Metricon system. Therefore, the values of TOC are apparently for the viscous state of the polymer.

Table 2 shows the refractive index of developed polymer films for various UV LED exposure rates along with their thermo-optic coefficients and estimated coefficients of thermal expansion using a Lorentz–Lorentz expression [7,32]. As underlined by Jang and Do (2014) [6], the refractive index decreases with the temperature increase as a result of thermal expansion. Derived values of coefficients of thermal expansion for formulations reveal differences as compared with the reference, as well as a decrease in their mean values, from 5.877 ± 0.020 (∙10^−4^) K^−1^ to 5.812 ± 0.106 (∙10^−4^) K^−1^ and 5.782 ± 0.045 (∙10^−4^) K^−1^, respectively.

#### 3.3.3. Optical Loss

As mentioned, a prism coupler system from Metricon was deployed to obtain the optical loss of the UV LED-cured fluorinated acrylates by the exponential fitting of the curve of light intensity scattered along the thin-film surface. Two wavelengths were used for the measurements, 633 nm (visible light range) and 1547 nm (near infrared), respectively. Appendix A exhibits excerpts of fitting for scattered light intensity variation function of distance from the coupling point as deployed further for calculation of optical loss. The selection was driven by requirements imposed on optical materials for use in fiber optics and waveguides.

Table 3 exhibits optical loss values of UV LED fluorinated acrylates as measured on the spin-coated formulations. As seen, different silane-based conditioners significantly increase or decrease this optical parameter, depending on the wavelength selected to run the measurements. As generally acknowledged, absorption and scattering represent the main factors influencing optical loss in polymers [39]. The highly fluorinated UV LED curable acrylates developed can be regarded to exhibit low absorption due to the presence of C=F bonds. The presence of this bond in a polymer chain was proven to reduce optical loss and increase thermal stability [40,41]. Consequently, scattering effects can be regarded to dominate this optical parameter presumably originating from inhomogeneities encountered at a lower scale or variations of the refractive index [34].

## 4. Conclusions

In this contribution, a series of perfluoropolyether (PFPE)-urethane methacrylate transparent coatings has been developed based on an acrylic mixture using a neutral fluorinated polymer network, i.e., Fluorolink^®^ MD 700 and cured, in dynamic mode, by means of a UV LED in-place developed conveyor belt configuration. Resulted coatings properties are highly influenced both by crosslinking agents and curing times.

The viscoelastic properties of transparent coatings developed, based on different crosslinking agents, showed degradation in their storage modulus and negative shifts in their T_g_ with respect to the selected reference. The reported experimental data indicate the combination using dimethyl(divinyl)silane as a crosslinking agent shows relatively small differences in viscoelastic properties compared to the reference, and therefore improved performance in different environments and circumstances. 

Refractive indices, independent of the crosslinking agent used, showed negligible variations with exposure rates or temperature increase compared to the reference coatings, with typical values of about 1.38 to 1.39. As for the thermo-optical coefficient (TOC), deploying trimethoxy(vinyl)silane as a crosslinking agent, result in a relatively small degradation in property compared to its counterpart, from −2.57∙10^−4^ K^−1^ to −2.55∙10^−4^ K^−1^. In addition, optical loss values for combinations showed different behavior at various wavelengths selected for measurements in favor of dimethyl(divinyl)silane while used as a crosslinking agent.

Consequently, the transparent perfluoropolyether (PFPE)-urethane methacrylate coatings described herein exhibit viscoelastic and optical properties that make them suitable for demanding applications requiring a low refractive index cladding for optical fibers or polymeric waveguides with low optical losses. The experimentally measured optical properties obviously support the combination with dimethyl(divinyl)silane as a crosslinking agent, although the other proposal should not be disclosed. Further applications can be identified with automotive engineering such as clear coats and coatings for battery elements, braking systems, optoelectronics, illumination sources, etc., as the demands evolve exponentially. To expand the application potentials, in addition to those already developed for aerospace, electronics and electrical industry, chemical processing, or medical applications, one should pay attention to the disruptive technology of 3D printing. With respect to the latter, these fluorinated polymers can be used as release agents or lubricants in the printing process to enable part removal and improve the printing quality. 

## Data Availability

The data presented in this study are available on request either from the first or corresponding authors.

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
