# Peer review of "UV LED Curable Perfluoropolyether (PFPE)-Urethane Methacrylate Transparent Coatings for Photonic Applications: Synthesis and Characterization"

_polymers, 2023, doi:10.3390/polym15142983_

Round 1
Reviewer 1 Report
In my opinion, the manuscript presents an interesting study of photo-curing properties of fluorinated materials. However, there are some points that should be addressed:
- What would be the chemical structure of the prepared system? Probably a reaction scheme would contribute to greater clarity
- It is not clearly established what is expected or the results obtained using two crosslinkers. Perhaps the variation in the concentration of one or the other would have been a parameter to consider
- Authors should comment on a specific application for the material
Author Response
Answers with the attached file.

Reviewer 2 Report
The current paper deals with the photopolymerization of UV-curable components. The paper is interesting and credible. The authors mention several contributions to the received results. While plenty of similar papers on the formulation are available using the same or other components; for example, paper 10.3390/polym13081195, which can be referenced if appropriate. The novelty of the chosen approach needs to be more justified. Additional data analysis is also suggested. Sol-gel and or FTIR need to be added to be sure that the developed resin compositions were UV-cured completely. FTIR methodology to evaluate the double bond conversion is available in 10.3390/polym13081195 Please also make a correlation between the crosslinking degree of the polymer and the optical properties depending on processing conditions.
Author Response
Answers with the attached file.

Reviewer 3 Report
Article is well prepared. Experiments are presented clearly. It could be published after minor revisions:
1. In Fig. 5 accuracy of the refractive index is really wondering. Especially, that in Fig 6 accuracy is 10 times lower. Similar in Tab 2.
2. Could you provode example FTIR spectra?
3. Line 295. Probably there is some mistake with values along with uncertainties. It should be 5.877∙10-4 ± 0.020∙10-4 K-1.
4. Line 306. Will be better to change (visible spectrum) into (visible light range).
5. Line 337. You provide values with x (-2.57x10-4 K-1 to -2.55x10-4 K), while in line 295 is with ".".
no comments
Author Response
Answers to the attached file.

Round 2
Reviewer 2 Report
The authors did several changes, while full FTIR spectra and discussion are missing. FTIR can be in Supplementary.
The novelty of the chosen approach needs to be more justified.
The novelty is not justified in the text.
Sol-gel
No additional data on sol-gel are provided, which is important for such a system.
Please also make a correlation between the crosslinking degree of the polymer and the optical properties depending on processing conditions.
No additional discussion is provided.
Author Response
Dear esteemed reviewer,
Answers to comments can be found in the attached version and traced with the newest version of the manuscript.
Thank you very much for your helping hand on improving our contribution.

Round 3
Reviewer 2 Report
-